# Examining the Canadian 24-Hour Movement Guidelines among Adults with Intellectual Disability: A Pilot Study

**DOI:** 10.3390/ijerph20136291

**Published:** 2023-07-04

**Authors:** John Cooper Coats, Matthew Coxon, Viviene A. Temple, Cara Butler, Lynneth Stuart-Hill

**Affiliations:** School of Exercise Science, Physical & Health Education, University of Victoria, Victoria, BC V8P 5C2, Canada

**Keywords:** intellectual disability, physical activity, sedentary behaviour, sleep, heart rate

## Abstract

The purpose of this pilot study was to investigate the extent to which adults with intellectual disability (ID) met the 2020 Canadian 24-Hour Movement Guidelines. Fifteen adults (six females and nine males) participated in this nine-day observational study (age = 20–64 years) in 2021–2022, during the COVID-19 pandemic. Moderate-to-vigorous physical activity (MVPA), sedentary time, and total sleep time were measured with a smartwatch to compare to the guidelines. A diary subjectively tracked physical activity. Of the 15 participants, 11 met the MVPA guidelines (73%), 4 met the sedentary behaviour guidelines (27%), 7 met the sleep guidelines (47%), and only 1 participant met all 3 of the guidelines (7%). There were no differences in physical activity or sleep between weekends and weekdays, or between males and females. Walking, cleaning dishes, and swimming were the most common types of physical activity performed by the participants. The findings of this pilot study indicate the need to improve sleep and reduce sedentary time in adults with ID. As most participants met the MVPA guidelines, few met the sedentary behaviour guidelines, and nearly half met the sleep guidelines, these data also demonstrate how important it is to assess all three aspects of the movement guidelines. All these behaviours have independent health benefits and risks, which interact to influence overall health.

## 1. Introduction

The hallmarks of a healthy lifestyle include maintaining regular physical activity, limiting sedentary time, and maintaining proper sleep hygiene. Maintaining a physically active lifestyle has both immediate and long-lasting benefits. A single bout of physical activity has been shown to reduce blood pressure and feelings of anxiety, as well as to improve sleep quality, insulin sensitivity, and cognitive function in the prefrontal cortex [1]. Maintaining regular physical activity also has long-term benefits, including reducing the risk of dementia, cardiovascular disease, stroke, several cancers, and weight gain, as well as improving overall bone health, balance, and coordination [1]. Meanwhile, inactivity and sedentary behaviour have been linked to an increased risk of cardiovascular disease, type II diabetes, and several cancers [1]. Maintaining healthy sleep hygiene has also been shown to reduce the risk of heart disease and type II diabetes, as well as to improve immune function, mood, and cognitive function [2].

The Canadian Society for Exercise Physiology (CSEP) published the 2020 Canadian 24-Hour Movement Guidelines, which describe the suggested amounts of physical activity, sedentary activity, and sleep to maintain a healthy lifestyle and lower the risk of chronic disease [3]. These guidelines suggest achieving a minimum of 150 min of MVPA each week, including two muscle-strengthening activities using major muscle groups [3]. Limiting sedentary time to eight hours or less and limiting recreational screen time to three hours or less are also recommended [3]. The guidelines also recommend achieving seven to nine hours of good-quality sleep regularly, with consistent rest- and wake-up times.

While the benefits of maintaining a healthy lifestyle are clear, there is growing evidence suggesting that adults with ID are not achieving healthy levels of physical activity, leading to poor sleep quality and detrimental health outcomes [4,5]. Furthermore, the COVID-19 pandemic provided additional barriers to maintaining a healthy and physically active lifestyle, which has disproportionately affected individuals with ID [6]. 

Due to the restrictions necessitated by the global pandemic, in-person data acquisition for research on physical activity, sedentary behaviour, and sleep among vulnerable populations became increasingly difficult. However, improvements in wearable technologies provided a unique opportunity for tracking physical activity, sedentary behaviour, and sleep in vulnerable populations [7]. Along with the physiological consequences of physical inactivity, there is a significant economic cost associated with the management and treatment of related diseases [8]. Therefore, it is of utmost importance that policymakers make informed decisions concerning efforts to mitigate the risk of chronic disease associated with physical inactivity among Canadians with ID backed with a strong evidence base.

Therefore, the objective of this pilot study was to determine the extent to which adults with ID in Victoria, British Columbia (BC), Canada were meeting the physical activity, sedentary activity, and sleep recommendations of the CSEP’s 2020 Canadian 24-Hour Movement Guidelines during the COVID-19 global pandemic [9]. The research questions were as follows. (1) To what extent are adults with ID in Victoria, BC meeting the Canadian 24-Hour Movement Guidelines during COVID-19? (2) What are the most common modalities for achieving physical activity among adults with ID? 

## 2. Materials and Methods

### 2.1. Participants and Recruitment Procedures

The University of Victoria’s Human Research Ethics Board approved the ethics application (protocol #20-0601) for this project prior to the recruitment of participants. The research team consisted of two professors, two graduate students, and two undergraduate students. All members of the research team completed the Canadian Tri-Council Policy Statement: Ethical Conduct for Research Involving Humans (TCPS-2) course prior to the start of data collection. Participants were recruited from several community organizations providing programming and support to adults with ID in Victoria, BC. ID is a disability characterized by deficits in both intellectual and adaptive functioning in the conceptual, social, and practical domains. Posters were put up in the organizations to directly recruit participants, and recruitment letters were emailed to potential participants directly or to their caregivers. Participants were recruited using convenience and snowball sampling. Eligible participants were: (1) 18 to 64 years of age; (2) a member of a community program for individuals with ID; (3) able to perform MVPA; and (4) willing to wear a Polar Ignite smartwatch [10] for nine consecutive days. Participants were excluded from the project if they: (1) had major mobility restrictions that prevented them from performing physical activity; or (2) were unable to wear the smartwatch for nine consecutive days. Volunteers who were interested and met the eligibility criteria were sent informed consent and assent letters that were signed prior to the beginning of data collection. Fourteen participants were able to provide written or verbal examples of their physical activity throughout the week, while one participant was non-verbal and required the assistance of a caregiver.

### 2.2. Study Design

The project was a pilot study that followed an observational cross-sectional design, and the participants were asked to carry on their normal daily activities. Data were collected by the participants with a Polar Ignite smartwatch (Polar Electro Oy, Kempele, Finland) and a physical activity and sleep diary. The participants were provided with a smartwatch to wear for nine consecutive days, as well as with a physical activity and sleep diary that was completed upon waking and before bed each day. This pilot study is part of a broader study that examined each of the three components of the 2020 Canadian 24-Hour Movement Guidelines in depth [9]. The present study focuses primarily on the physical activity and sedentary behaviour aspects of the guidelines.

### 2.3. Instruments and Measures

#### 2.3.1. Participant Demographic Characteristics

The participants were measured for height (metres) and weight (kg) at the beginning of the data collection, and their Body Mass Index (BMI) was calculated. Height was measured by having the participants stand without shoes beside a wall lined with a Stanley Powerlock Tape Measure (product #33-430). Weight was measured by having the participants stand without shoes on a Philips Body Analysis Scale (product #DL8781/38), which rested on a flat surface. Age, sex, Down syndrome status, and whether the individual had a major mobility restriction that prevented them from completing physical activity were identified by the participant or their caregiver.

#### 2.3.2. Polar Ignite Smartwatch

The main instrument used for the pilot study was the Polar Ignite smartwatch with integrated Global Positioning System (GPS) and wrist-based pulse rate [10]. It is a small (dimensions: 43 × 43 × 8.5 mm) and lightweight (weight: 35 g with wristband) device with GPS, Bluetooth, and heart rate monitoring capabilities. The device is worn on the wrist of the non-dominant hand, with the device fitting snugly but comfortably to the wrist. For data collection purposes, the “Other Indoor” mode on the device was used, which constantly tracked the participants’ heart rate (1 s averages) and total sleep length with interruptions. Previous research has demonstrated that heart rate data from the Polar Ignite smartwatch showed strong correlation values (*r* > 0.90) when compared to the Polar H10 chest strap heart rate monitor during exercise [11]. Furthermore, a similar device by Polar, the A370 fitness watch [12], showed no significant difference when compared to polysomnography in terms of measuring the time between sleep onset and offset (i.e., falling asleep and waking up) in both young and older age groups (*p* = 0.10 and 0.46) [13]. The minimum wear time criterion for inclusion in the study was 4 days of 24 h. 

#### 2.3.3. FlowSync App and Polar Flow

All the data collected from the Polar Ignite smartwatches were uploaded to the FlowSync application (version 6.7.0, Polar Electro Oy, Kempele, Finland) on a Samsung Tablet for storage purposes [14]. The FlowSync app tracks and organizes all heart rate data (measured in 1 s averages) and all sleep data (measured in minutes). Once uploaded to FlowSync and connected to the internet, the data were automatically synced to the online Polar Flow software (Polar Electro Oy, Kempele, Finland). The Polar Flow software organized each data collection session (using the “Other Indoor” mode) in a training calendar for further analysis. Upon selection of a data collection session, the software chronologically displayed the activity of the individual wearing the smartwatch, including the duration of the session, heart rate data (1 s averages, total average, minimum, and maximum), kilocalories burned, as well as sleep length and interruptions.

#### 2.3.4. Physical Activity, Sedentary Behaviour, and Sleep

Objective Measures: To compare to the recommended guidelines, heart rate data (1s averages) from the Polar Flow software were used to extrapolate the time spent performing physical activity (MVPA and light physical activity [LPA]) and sedentary behaviour. While the total sleep time was taken from the sleep data in the Polar Flow software, the MVPA and LPA were extrapolated from the heart rate data by summating the amount of time spent with a heart rate greater than or equal to 64% of the heart rate maximum (HRmax) and 50–64% of the HRmax, respectively [15]. The maximal heart rate was estimated using Fernhall et al.’s [16] formula specific for persons with Down syndrome: HR_max_ = 210 − (0.56 × Age) − (15.5 × DS), age = years, DS: 1 = diagnosis of Down syndrome, 0 = diagnosis of intellectual disability but not Down syndrome. Sedentary behaviour was extrapolated from the heart rate data by summating the amount of time spent with a heart rate ≤50% of the HRmax using the same methodology as above [15]. Data were collected from the participants for 9 consecutive days, and the data was transformed into a 7-day average using daily amounts to standardize the data sets. Lastly, the total sleep length was taken from the Polar Flow software and used as a metric to compare with the recommended guideline of 7 to 9 h per night [9].

Subjective Measures: The physical activity and sleep diary was used to collect data on the number of bouts and types of physical activity and information about sleep. Sleep was explored in greater detail in the broader study, while only the total sleep time was used in this pilot study. The sleep portion of the diary was based on the Sleep Foundation’s diary [17]. The physical activity section of the diary used in this pilot study was adapted from Temple and Walkley [18]. Each participant was instructed to complete the physical activity section of the diary before getting ready for bed each day. The physical activity section had a list of YES or NO questions pertaining to various forms of activity performed (e.g., sports, non-sport activities, household chores, and yard work) with examples (e.g., washing dishes for a household chore) for each of the 8 days in the diary. There were 8 days included in the diaries to ensure at least 7 full days of activity were collected to compare with the objective measures. There was also some additional room to provide extra information where applicable (e.g., soccer for sport played). Once the diary was complete, all the data were organized in an Excel document. For each question, the participant’s responses were tabulated and a weekly total number of YES responses was determined (max. 8). For each YES response, the example of the activity (e.g., walk) was also recorded to determine the most common types of activity. Once completed, the number of bouts of physical activity of all types (sport-related, non-sport-related, household, yard work) were measured. As the intent of the physical activity diary was to establish what activities the participants were doing rather than the actual minutes of MVPA or energy expenditure, the validity of the responses was confirmed during the “check-ins” by asking the participants about those activities.

### 2.4. Procedure

Data were collected from each participant for nine days to ensure a total of seven uninterrupted days were included in each data set. On the first day, a member of the research team arrived at the community program centre or predetermined location and introduced them self to the eligible participant while following the COVID-19 safety protocols by wearing a mask and using disposable laboratory gloves. The researcher then collected height and weight measurements as well as demographic information from the participant (age and sex). Down syndrome status and any major mobility restriction were identified by the participant or caregiver prior to the start of the data collection. Once collected, the participant was fitted with the Polar Ignite smartwatch and the participant was reminded of the data collection protocol. Specifically, a member of the research team would return to collect the data onto the Samsung tablet and charge the device every second day. The participant was familiarized with the Polar Ignite smartwatch and practiced putting the smartwatch on and taking it off. The participant and caregiver (if necessary) were instructed on how to charge the device should it run out of battery, and this was practiced. The participant was also familiarized with how to complete the physical activity and sleep diary. Once the participant was instructed on how to use the diary, the researcher asked the participant if they could provide an example for each section to confirm their understanding of the task. The data collection schedule is listed in Table 1.

After the participant was familiarized with the researcher, equipment, and data collection procedure, the researcher then verbally confirmed that the participant understood these components of the study. The researcher then initiated the “Other Indoor” mode on the Polar Ignite smartwatch and the nine-day protocol commenced. During the data collection period, the participant was followed up every second day by a member of the research team to ensure that the diary was being completed properly. During each follow-up session, the research team checked for missing data. In all cases regarding the diary, the participant or caregiver added the missing information. Nearly all the participants were able to provide verbal or written examples of their physical activity, while one participant required the assistance of a caregiver. Once the nine-day protocol was completed, the smartwatch and diary were retrieved from the participant and the data for the last two days were uploaded to the Polar Flow software. The smartwatch and the diary were stored in a locked cabinet located in the researchers’ laboratory. The participant was provided with a gift card as a small thank you at the end of the data collection period.

### 2.5. Data Treatment and Analysis

The participants’ information was uploaded to a secure Excel document on a password-protected computer and a participant number was assigned to the data set to anonymize the data. Height and weight data were used to determine the BMI for each participant using the formula kg/m^2^. 

Each Polar Ignite smartwatch had an identifier number, which corresponded to an anonymized participant number and a Polar Flow account. Data were stored on the password-protected Polar Flow application. The heart rate averages (1 s) for each data collection period were downloaded and collated in an Excel file for each participant. The participants’ data were standardized to a 7-day measure using the daily averages and the weekly MVPA, LPA, and sedentary time were calculated and converted into minutes. Each data set had a specific start time and date, and thus the amounts of MVPA, LPA, and sedentary time were determined for weekdays and weekends. Sleep data were uploaded and stored on the Polar Flow software and the total sleep time was extracted from the software. Once each participant had a weekly summary of their MVPA, LPA, sedentary time, and sleep, descriptive statistics were used to summarize the group’s data (mean, standard deviation, 95% CI). 

Using IBM SPSS Version 27.0 (IBM Corp, Armonk, NY, USA).and a significance level of 0.05 (α), Mann–Whitney U tests with two-tails were used and *p*-values were determined to compare the MVPA, LPA, and sedentary behaviour between weekdays and weekends, and between males and females.

## 3. Results

Table 2 shows the demographic characteristics of the 15 participants included in the analysis of this pilot study.

The weekly sedentary activity and MVPA and daily sleep time were determined for each participant, as shown in Table 3. The time spent sedentary ranged from 21.9 h to 147.6 h over 7 days, while the MVPA ranged from 26 min to 1509 min. The mean actual sleep time ranged from 4 h 43 min to 9 h 2 min. The numbers of participants who met each of the CSEP’s 24-Hour Movement Guidelines are shown in Table 3.

Of the 15 participants, only 1 met all 3 of the CSEP’s 24-Hour Movement Guidelines, and the modal score for meeting all three guidelines [9] was one (Table 3). MVPA was the most common guideline to have been met, while the sedentary activity and sleep guidelines were met the least by the participants (Table 3).

When comparing the physical activity during weekdays and weekends, there were no significant differences between the levels of sedentary activity, LPA, and MVPA, as determined by the two-tailed Mann–Whitney U tests. The effect sizes for the differences between weekdays and weekends were small [20], specifically d = 0.04, d = 0.06, and d = 0.22 for sedentary activity, LPA, and MVPA, respectively. Similarly, there were no significant differences between the male and female participants. The effect sizes of the comparisons between the males and females were d = 0.03, d = 0.59, and d = 0.09 for sedentary activity, LPA, and MVPA, respectively (Table 4).

The second research question focused on the modalities for achieving physical activity. The daily activity diaries were used to understand how the participants were getting their physical activity. As shown in Figure 1, walking, cleaning dishes, and swimming were the most common types of physical activity. Non-sport physical activity was the most common modality for physical activity, as 72.5% of days recorded for all the participants included at least one non-sport-related activity, while yard work was the least common, with 0.8% of days recorded. Sports were played on 15.8% of the recorded days and household activity was conducted on 52.5%. 

## 4. Discussion

Maintaining a physically active lifestyle and reducing the amount of time spent sedentary has been shown to promote overall well-being and reduce the risk of developing chronic disease [1]. Furthermore, achieving a healthy level of 7–8 h of sleep has also been shown to provide a plethora of physiological and psychological benefits [21]. To promote a healthy lifestyle that focuses on promoting physical activity and sleep while limiting the time spent sedentary, the CSEP released the 2020 24-Hour Movement Guidelines, which include recommendations for all stages of life [3]. Yet, despite the known benefits of maintaining a healthy lifestyle, there is an increased prevalence of physical inactivity, sedentary activity, and poor sleep quality reported among adults with ID [4,5]. Physical inactivity has been linked to health detriments, including cardiovascular disease and obesity [22]. Furthermore, the COVID-19 global pandemic prevented adults with ID from accessing community programs and facilities during lockdowns, which exacerbated these issues [6]. Therefore, there was a need to explore the ways through which individuals with ID were achieving physical activity and getting enough sleep during COVID-19. 

The aim of this pilot study was to determine the extent to which adults with ID were meeting the physical activity, sedentary activity, and sleep recommendations of the CSEP’s 2020 Canadian 24-Hour Movement Guidelines during the COVID-19 global pandemic [9]. This study also piloted the use of commercially available wearable technology and a diary to investigate how the participants were achieving physical activity while attempting to reduce the overall contact time between the researchers and participants.

The Canadian 24-Hour Movement Guidelines [9] recommend that adults engage in a minimum of 150 min of MVPA weekly. A total of 11 of the 15 participants (73%) met this recommendation. This finding was unexpected, as previous studies using similar physical activity criteria demonstrated that only 9% of adults with ID achieved 150 min of MVPA weekly [4]. The limited literature on the physical activity of individuals with ID during the pandemic generally showed a negative effect. On the other hand, some organizations, such as the Special Olympics, made considerable efforts to help individuals with ID stay active during the pandemic by providing virtual sports and fitness programming as well as return to activities guidelines [23]. Furthermore, previous work in this community has revealed considerable variation in the physical activity levels of adults with ID [24], including a proportion of very active participants [25]. However, the exact reasons for why this sample of participants was highly active are unknown.

Far fewer participants (27%) met the sedentary behaviour recommendation of less than 8 h per day. This finding aligns more closely with contemporary research, as sedentary behaviour is commonly reported among adults with ID and is of concern for both physiological and psychological health [4,26]. It should be noted, however, that the average weekly time spent in LPA in the present study was 2804 min. Although LPA is not a measure included in the CSEP movement guidelines, LPA is favourably associated with a lower overall mortality risk and improved health outcomes [27]. As mentioned previously, sedentary behaviour is common in this population and may have been exacerbated by the global pandemic restrictions [6].

The final recommendation of the Canadian 24-Hour Movement Guidelines is achieving seven to nine hours of quality sleep. Using the total sleep length as a measure of sleep quality, we found that 7 of the 15 individuals (47%) met the minimum recommendation. Although the mean sleep time for all the participants was 6 h and 55 min, there was a considerable range of 4 h and 43 min to 9 h and 2 min. It is generally understood that sleep disruptions are common among adults with ID, as shown in one meta-analysis by Surtees et al. [28], who found that 93% of comparisons between individuals with and without ID indicated poorer sleep quality in those individuals with ID. The finding of this pilot study does not fully align with these previous studies, as nearly half of the participants were able to maintain an adequate 7 h of sleep. Overall, when combining all three aspects of the Canadian 24-Hour Movement Guidelines (MVPA, sedentary behaviour, and sleep), there was only one participant who met all the guidelines. 

The COVID-19 global pandemic brought about a significant reduction in access to community gyms and facilities [29]. Individuals with ID were particularly vulnerable to the disruption, and contemporary research has indicated a significant reduction in physical activity and greater sedentary behaviour [6]. Consistent with this previous research, the present study found that only 27% of participants met the recommendation for sedentary behaviour. However, the present study also found that 73% of participants were able to meet the recommendation for MVPA despite the restrictions of COVID-19. These findings are consistent with a phenomenon colloquially referred to as the “Active Couch Potato.” This phenomenon occurs when individuals are achieving a healthy level of MVPA but are spending too much time sedentary when not exercising [30]. Despite the known reduction in the risk of developing chronic disease from maintaining a healthy level of physical activity, there is a growing body of research indicating an independent, dose–response relationship between sedentary behaviour and the risk of all-cause mortality [31]. Therefore, this pilot study’s findings indicate a real need to address the issue of sedentary behaviour in adults with ID. 

Future studies should look to build upon this pilot study by implementing a larger random sample over several weeks to gain a better understanding of the activity levels of participants over a typical week. Given the clear indication from this study that the participants were spending too much time sedentary, future studies should attempt to track screen time to compare it to guidelines (e.g., 8 h or less). The present study used only the total sleep time as a metric for sleep, although other work performed as part of the broader study examined sleep quality and consistency, as well as sleep hygiene behaviours, which should allow for a more comprehensive investigation of sleep.

The present study investigated how adults with ID achieve regular physical activity. Using the physical activity and sleep diary, data on the types and number of bouts of physical activity were collected over a week and examined for commonalities. Most previous studies that used both self-report and objective approaches to measure physical activity typically looked at walking/running and/or sport-related activity without a focus on household activities or chores [4]. Therefore, it was the focus of this study to include a section in the physical activity and sleep diary on household activities such as cleaning one’s room and taking out the garbage. The most common type of activity was non-sport-related activity (e.g., walking and lifting weights), followed by household activity (e.g., cleaning dishes and room), sport-related activity (e.g., swimming), and yard work (e.g., racking leaves) (Figure 1). More specifically, walking was the most common modality for physical activity in this study, aligning with the contemporary literature on adults with ID [32]. Cleaning dishes, an indoor household chore, was the second most reported activity completed by the participants in this study (Figure 1). Although not many studies on adults with ID have focused on chore-related work for achieving physical activity, one study by Barnes et al. [33] found that indoor chore-related work was the second most reported activity on self- or proxy-reported physical activity reports (42.5% of participants). However, an observational case study of physical activity behaviours found chores contributed only a small proportion of MVPA among six individuals living in a group home [34]. Nonetheless, this pilot study’s findings support the identification of all forms of moderate- and light-intensity physical activity in future studies, including household chores. Lastly, the third most reported activity in this study was swimming, which has been often reported among this population at all ages, as it is a low-impact activity that improves joint mobility and cardiovascular fitness [35,36]. As mentioned previously, access to community gyms and facilities was significantly affected during this period of the global pandemic, leading to a disproportionate decrease in physical activity among adults with ID [37]. These restrictions could potentially be a reason for the distribution of the types of physical activity, as activities such as lifting weights, yoga, and group dance all typically require the use of a community space, which was restricted during COVID-19.

A limitation of the current literature on physical activity in adults with ID is that very few studies focused on the modalities of physical activity used and rather focused on the number of minutes performed to be able to compare that to recommendations [4]. The present pilot study included a focus on the types of activities that adults with ID are using to achieve physical activity to investigate both the amount and type of physical activity performed throughout a typical week. Future studies should look to include information pertaining to the physical demand of a participant’s employment, which could allow for comparisons between working and off days. Lastly, since the 2020 Canadian 24-Hour Movement Guidelines specify that adults should be engaging in at least two sessions of muscle-strengthening activity (e.g., resistance exercise) per week, future studies should look to include a section in the self-report that highlights this recommendation [9].

The Polar Ignite smartwatches provided the present study with a technology that allowed for the remote monitoring of activity and sleep, as was crucial for the completion of the data collection during the restrictions of the COVID-19 global pandemic. During the present study, 11 of the 15 participants wore the Polar Ignite smartwatches for 7 or more days of data collection, 3 participants wore them for 6 days, and 1 participant wore it for 5 days. Of the days collected, the Polar Ignite smartwatches were able to collect data 89.2% of the time. These findings indicate that the watches were generally well-received and comfortable to wear. One issue with the current literature is that all the validation studies on devices such as the Polar Ignite smartwatch only included individuals without a disability or health condition, and therefore, there is a need for a more inclusive participant population in future validation studies [38,39]. Although there were minimal setbacks when using the Polar Ignite smartwatches during this pilot study, the next steps in this line of research must be to establish the construct and concurrent validity. These steps could include the use of concurrently wearing other accelerometers validated with this population and direct observation of behaviours [34,40].

## 5. Conclusions

We explored the extent to which a sample of adults with ID were meeting the 2020 Canadian 24-Hour Movement Guidelines [9], with a greater focus on physical activity and sedentary behaviour. Sleep was investigated in greater detail in other parts of the broader study. Our findings revealed that very few participants met the movement guidelines in their entirety. A large proportion of the participants in this study displayed characteristics of what has been colloquially called the “Active Couch Potato” phenomenon [30], where the participants achieved an adequate amount of MVPA but spent far too much time sedentary. Achieving a healthy level of physical activity has been shown to improve overall heart and brain health, physical fitness, and quality of life, as well as to lower the risk of all-cause mortality [1]. However, despite achieving enough MVPA, active individuals living a sedentary lifestyle are at a heightened risk of developing metabolic disorders and all-cause mortality [31]. Secondly, the participants did not achieve enough sleep, which is worrisome because lacking adequate sleep can increase the risk of cognitive disorders and decline [41,42], CHD [43], and type II diabetes [44]. Therefore, the findings of this pilot study indicate that there is a real need to further investigate and address the high level of sedentary behaviour and lack of sleep among adults with ID. An important insight from this study is that the participants were able to wear the watches all day and night for many days. The data from the 24 h cycles provide a more complete picture of several interrelated health behaviours. Achieving an adequate amount of physical activity, limiting sedentary behaviour, and maintaining good sleep hygiene are all incredibly important for overall health and longevity. It is therefore paramount that governments attempt to promote physical activity and sleep for individuals with ID through the support of community inclusion programs. 

## Figures and Tables

**Figure 1 ijerph-20-06291-f001:**
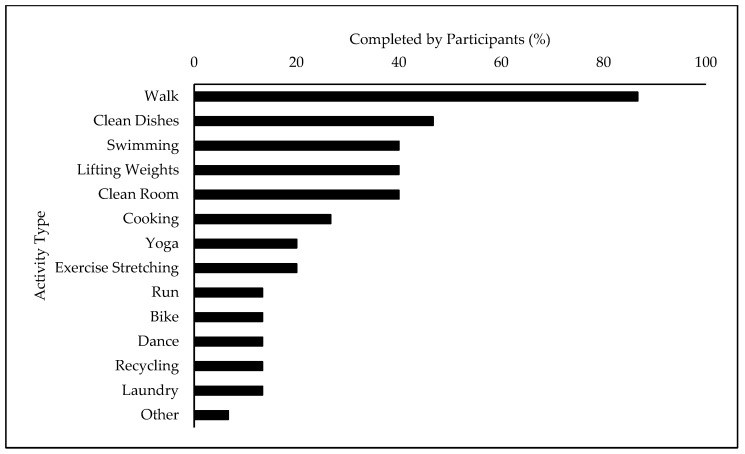
Most common types of physical activity performed during a week. The measures above refer to the percentage of participants who engaged in that activity at least once throughout the week for the purpose of attaining physical activity. The participants used their physical activity and sleep diaries to indicate what activity they performed each day.

**Table 1 ijerph-20-06291-t001:** Schedule for the Polar Ignite data collection protocol.

Day	Action
1	Demographic information collected, smartwatch and diary distributed
2	Remote monitoring
3	Data collected and smartwatches charged
4	Remote monitoring
5	Data collected and smartwatches charged
6	Remote monitoring
7	Data collected and smartwatches charged
8	Remote monitoring
9	Smartwatches and diary retrieved

Note. Demographic information included height, weight, and sex. Remote monitoring refers to when the participants were wearing the Polar Ignite smartwatch with active tracking of their pulse rate. The diary refers to the physical activity and sleep diary.

**Table 2 ijerph-20-06291-t002:** Descriptive statistics of the participants’ demographic characteristics (*n* = 15).

	Overall Sample *n* = 15			Male Participants *n* = 6			Female Participants *n* = 9		
**Metric Variables**	**Mean**	**SD**	**Range**	**Mean**	**SD**	**Range**	**Mean**	**SD**	**Range**
Age (years)	35.9	13.5	20–64	38.0	14.5	21–58	34.4	13.6	20–64
Height (cm)	160.1	12.7	140–187	169.5	11.1	152–187	153.8	9.6	140–168
Weight (kg)	72.1	17.2	45–104	80.2	13.6	55–95	66.7	17.9	45–104
**Categorical Variables**	** *n* **	**%**		** *n* **	**%**		** *n* **	**%**	
Weight status *									
Underweight	0	0		0	0		0	0	
Healthy weight	6	40.0		2	33.3		4	44.4	
Overweight	5	33.3		3	50.0		2	22.2	
Obese	4	26.7		1	16.7		3	33.3	
Down syndrome	3	20.0		1	16.7		2	22.2	

Note. * Based on the BMI categories outlined in the Canadian Guidelines for Body Weight Classification in Adults [19]. SD = standard deviation.

**Table 3 ijerph-20-06291-t003:** Mean amounts of physical activity, sedentary activity, total sleep time, and guidelines met.

Part. ID		Weekly MVPA (min)	MG	Weekly Sedentary Activity (h)	MG	Daily Sleep Time (h:min)	MG	Total MG Met
1	M	1283	✓	54.4	✓	7:26	✓	3/3 *
2	M	111		147.6		6:45		0/3
3	F	900	✓	80.5		6:14		1/3
4	F	1086	✓	77.9		7:42	✓	2/3
5	M	1509	✓	21.9	✓	5:41		2/3
6	M	165	✓	135.5		6:13		1/3
7	F	144		138.1		9:02	✓	1/3
8	F	26		51.7	✓	4:51		1/3
9	M	166	✓	78.5		7:26	✓	2/3
10	F	1179	✓	96.3		8:24	✓	2/3
11	F	529	✓	116.5		8:10	✓	2/3
12	F	905	✓	104.8		4:43		1/3
13	M	556	✓	114.6		6:30		1/3
14	F	106		49.1	✓	7:44	✓	2/3
15	F	362	✓	103.7		6:59		1/3
Mean		602		91.4		6:55		
SD		500		36.7		1:04		
Total MG			11/15		4/15		7/15	1/3 **

Note. MVPA = moderate-to-vigorous physical activity; sleep time refers to total actual sleep (without interruptions). For adults aged 18–64, the CSEP Guidelines suggest maintaining 150 min MVPA/week, limiting sedentary time to 8 h/day (56 h/week) or less, and getting 7 to 9 h of sleep [9]. M/F = male/female; MG = meeting guideline; SD = standard deviation; ✓= successfully met guideline. * = met all the guidelines. ** = mode.

**Table 4 ijerph-20-06291-t004:** Differences in sedentary and physical activity between weekdays and weekends and between males and females.

Activity Type	Weekday (min/day)	Weekend (min/day)	*p*-Value (α = 0.05)
Sedentary Activity	794.07 ± 321.04	781.55 ± 298.27	1.000
LPA	399.40 ± 272.78	414.14 ± 260.51	0.838
MVPA	92.02 ± 77.480	77.57 ± 52.57	1.000
**Activity Type**	**Male (min/week)** ** *n = 6* **	**Female (min/week)** ** *n = 9* **	** *p* ** **-Value (α = 0.05)**
Sedentary Activity	5524.62 ± 2940.93	5457.72 ± 1753.83	0.864
LPA	3496.54 ± 2468.43	2342.22 ± 1281.40	0.388
MVPA	631.81 ± 617.050	581.81 ± 446.32	0.689

Note. Data values are mean ± SD. *p*-values for comparisons were determined by two-tailed Mann–Whitney U tests using an alpha of 0.05. LPA = light physical activity; MVPA = moderate-to-vigorous physical activity.

## Data Availability

The data from this study are available from the corresponding author upon request.

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
