# Peer review of "Examining the Canadian 24-Hour Movement Guidelines among Adults with Intellectual Disability: A Pilot Study"

_ijerph, 2023, doi:10.3390/ijerph20136291_

Round 1
Author Response
Dear Reviewers,
Thank you for taking the time to review our submission diligently, we genuinely appreciate your comments. We have gone through your suggestions and have made changes accordingly, again thank you for improving our submission. Please find the attached updated manuscript and responses to each comment.
Sincerely,
John Cooper Coats

Reviewer 2 Report
Dear Authors,
I read with interest the manuscript received for evaluation. Your research summarizes the important issues recently published by the first author: (http://dspace.library.uvic.ca/bitstream/handle/1828/15064/Coats_John%20Cooper_MSc_2023.pdf?sequence=1&isAllowed=y). The research is useful through the studied topic, the manuscript is studied logically, the purpose of the research and the two investigated directions are clearly specified at the end of the introduction. You have identified the need to combine 3 different factors to favorably influence health and longevity (physical activity, decrease in sedentary behaviors and sleep quality). I have a few questions and can suggest some ideas for improving the initial version of the reviewed study:
1. What is the category/severity levels of classifying the investigated subjects for Intellectual Disability (mild, moderate, severe, profound)?
2. Why is the investigated group so small (only 15 participants/6 men and 9 women)?
3. It would be useful to also present details related to the job of each participant (if there are physical demands and at what level.....)
4. You used parametric statistical tests (t-test) to compare results between independent samples (men vs. women and weekdays vs. weekends). Have you applied tests to assess the normality of the distribution of results (Shapiro–Wilk test)? If the data are not normally distributed, then it would have been useful to apply non-parametric tests for independent samples (Mann–Whitney U).
5. Table 2: I think it would be useful to present the data separately by gender and at the end their sum.
6. Table 3: Participant number 2 is missing/numbering is wrong (there are 15, not 16 participants). You could also feature the gender of each participant.
7. Line 245-251 (independent samples t-test comparison results). Their presentation in tables (with mean values, SD, differences between means, t-values and Sig.) would make the text clearer to understand (in the work mentioned in the first paragraph they are included).
8. Figure 1: Swimming ranks third. Did participants have access to this type of physical activity during the pandemic? A gender comparison of these activities would also be useful.....there may be differences between them. Example: Lifting Weights may show higher scores for men.
9. Certain references are excessively cited, even though it is obvious that they are very important to your research: 1,3,9,10, 14, etc. Example: source 10 (lines 82, 90, 109, 111, 114, 115, 132, 179, 182, 209, 379, 382, 384, 387).
Author Response

(The authors gave the same response as above.)
